# Biologically-Based Notions About Uterine Bleeding During Myomectomy: Reasoning on Tradition and New Concepts

**DOI:** 10.3390/medsci13020068

**Published:** 2025-06-01

**Authors:** Andrea Tinelli, Giovanni Pecorella, Gaetano Panese, Andrea Morciano, Antonio Malvasi, Mykhailo Medvediev, Safak Hatirnaz, Radmila Sparic, Michael Stark

**Affiliations:** 1Department of Obstetrics and Gynecology and CERICSAL (CEntro di RIcerca Clinico SALentino), Veris delli Ponti Hospital Scorrano, 73020 Lecce, Italy; giovannipecorella2690@gmail.com (G.P.); gaetano.panese@gmail.com (G.P.); 2Department of Gynecology and Obstetrics, Panico Pelvic Floor Center, Pia Fondazione “Cardinale G. Panico”, 73039 Tricase, Italy; drmorciano@gmail.com; 3Obstetrics and Gynecology Unit, Department of Biomedical Sciences and Human Oncology, University of Bari “Aldo Moro”, 70121 Bari, Italy; antoniomalvasi@gmail.com; 4Department of Obstetrics and Gynecology, Dnipro State Medical University, 49044 Dnipro, Ukraine; medvedev.mv@gmail.com; 5Mediliv Medical Center, Samsun 55020, Turkey; safakhatirnaz@gmail.com; 6Clinic for Gynecology and Obstetrics, University Clinical Center of Serbia, Faculty of Medicine, University of Belgrade, 11000 Belgrade, Serbia; 7New European Surgical Academy (NESA), 10117 Berlin, Germany; mstark@nesacademy.org

**Keywords:** uterine fibroids, intracapsular myomectomy, surgical bleeding, myoma pseudocapsule, uterine artery clamp, embolization, Torniquet, vasopressin, tranexamic acid, uterotonics

## Abstract

Uterine fibroids represent a prevalent category of tumors encountered in females of reproductive age, may present as singular or multiple entities and can manifest a variety of symptoms, which can negatively affect women’s daily lives. Pharmacological interventions may prove to be ineffective, occasionally costly, and associated with adverse effects. In instances where symptoms escalate in severity, myomectomy becomes a requisite as uterine-preserving operative therapy. Myomectomy can be performed utilizing laparoscopic, robotic, laparotomic, vaginal or hysteroscopic techniques. Given the abundant vascular supply to the myometrium, with blood being delivered to the uterus via the uterine arteries, myomectomy carries a considerable risk of significant hemorrhage during and subsequent to the surgical procedure, with the related complications. This paper aims to elucidate the conventional methodologies employed to mitigate hemorrhage during myomectomy and in the immediate postoperative phase, evaluating the effect of chemical interventions (such as vasopressin, octreotide, tranexamic acid, and uterotonics) alongside mechanical strategies (including uterine artery clamps, embolization, and tourniquets) to curtail bleeding during the myomectomy process. Furthermore, the potential of employing the intracapsular myomectomy technique without reliance on other traditional approaches was explored. This surgical method is grounded in the principles of the biological and anatomical characteristics of the fibroid, facilitating the enucleation of the myoma from its pseudocapsule. This anatomical entity, which is formed by the myoma throughout its development within the myometrium, enables the fibroid to be detached from the uterine musculature and supplies the requisite neurovascular support for its sustenance. Finally, the narrative review also shows how the intracapsular approach, which uses the fibroid’s biology, reduces bleeding during myomectomy.

## 1. Introduction

Uterine fibroids constitute the most prevalent benign solid neoplasms found within the female demographic, arising from the smooth muscle tissue of the uterus, with their growth being facilitated by estrogen and progesterone. The prevalence of these fibroids among pre-menopausal women fluctuates between 20% and 40%; however, it is important to highlight that approximately 70% of cases remain asymptomatic and are identified incidentally during radiological examinations. Symptomatic uterine fibroids can manifest a variety of clinical symptoms based on their size and anatomical location, including bulk-related symptoms (such as pelvic pressure, lumbar or abdominal discomfort, and a sensation of fullness), dysmenorrhea, urinary or gastrointestinal issues, and sexual dysfunction; furthermore, they may also be associated with infertility or adverse obstetric outcomes, which encompass an increased risk of preterm labor, cesarean delivery, antepartum hemorrhage, abnormal fetal positioning, and intrauterine growth restriction [1]. The detrimental effects of these fibroids on the quality of life are considerable, as they lead to reduced work productivity and disruptions in both social and familial engagements. Factors such as the dimensions and positioning of fibroids, the patient’s age, symptomatic presentation, desire to maintain fertility, and accessibility to therapeutic interventions should be thoroughly evaluated when devising a management plan for fibroids. Numerous pharmacological agents, including chemical therapies, can be employed in medical management to alleviate heavy menstrual bleeding. Certain medications are viable options for patients requiring symptom relief prior to surgical intervention or those approaching menopause. This paper predominantly concentrates on myomectomy, the most frequently performed surgical operation, which can be executed through laparoscopic, robotic, laparotomic, vaginal, or hysteroscopic methodologies. The objective of this manuscript is not to critique the quality of the studies, sample sizes, or any discrepancies or biases present among various studies regarding hemorrhaging during myomectomy. By contrasting the advantages of chemical interventions (including vasopressin, octreotide, tranexamic acid, and uterotonics) with mechanical methods (such as uterine artery clamps, embolization, and tourniquets) aimed at mitigating bleeding during myomectomy, this paper endeavors to elucidate the standard techniques employed to minimize hemorrhage during the procedure and in the immediate postoperative phase. Furthermore, given that the intracapsular myomectomy technique is predicated on the biological and anatomical characteristics of the fibroid and facilitates the removal of the myoma from its pseudocapsule, the authors explore, in this narrative review, the feasibility of utilizing this technique independently of other conventional approaches in all instances of fibroid excision from the uterus.

## 2. The Main Procedure for the Removal of Intramural and Subserosal Fibroids: The Myomectomy

The abdominal myomectomy is a type of invasive surgery that can be performed robotically, laparoscopically or traditionally, by laparotomy. The first myomectomy was performed by Washington and John Atlee in 1844, and it has since established itself as the standard operation when the preservation of fertility and the uterus is considered vital [2]. Nevertheless, excessive bleeding, pyrexia, visceral injury, thrombosis, conversion to hysterectomy, blood transfusions, scar dehiscence in subsequent pregnancies, and several other intraoperative and postoperative problems are linked to abdominal myomectomy (Figure 1).

Significant blood loss from uncontrolled bleeding may cause hemodynamic instability and, in extreme situations, even a potentially fatal hemorrhage. Sometimes, a conversion to laparotomy is required to guarantee adequate hemostasis and prevent severe problems during myomectomy [3].

Although the amount of blood lost after myomectomy procedures varies, women with large or many fibroids may experience an average blood loss of up to 800 mL [4,5,6].

But, on the other side, the option of uterine resection is frequently deemed unacceptable for women desiring to conceive and may lead to deleterious psychological repercussions. Indeed, of the over 500,000 hysterectomies executed annually for benign conditions, more than one-third are attributable to symptomatic fibroids [7].

Furthermore, with the progression of assisted reproductive technology (ART), the preservation of the uterus has gained increasing significance. The emergence of minimally invasive surgery (MIS), including laparoscopic, robotic-assisted and hysteroscopic myomectomy, has consequently gained broader acceptance. These MIS approaches have progressively established themselves as the predominant techniques within the surgical theater, surpassing traditional laparotomic myomectomy [8].

A recent review compared laparoscopic myomectomy with laparotomic myomectomy, concluded that Hospital stays [SMD 3.12, IC 95% 0.57 to 4.28] and estimated blood loss [standard mean differences (SMD) 0.72, IC 95% 0.22 to 1.22] were considerably lower in the laparoscopic group than in the open group. The frequencies of pregnancy, intraoperative and postoperative complications, and other obstetrical outcomes did not differ statistically significantly between the two surgical techniques [9]. According to the results of the study, laparoscopic myomectomy showed several advantages over abdominal myomectomy, such as less blood loss, shorter hospital stays, and a lower need for postoperative analgesics. It also has comparable obstetrical outcomes and no appreciable increase in complication rates. However, the limited number of randomized trials on the selected group may limit the generalizability of these findings. Furthermore, the gynecologist’s skill had a direct impact on patient selection, whereas the healthcare facility’s policies had an indirect effect [9].

## 3. The Problem of Blood Loss During Myomectomy

Infertility and fibroids in older patients, as well as gynecologists who have unwarranted confidence in myoma excision by MIS without adequate training in laparoscopic myomectomy and myometrial suturing, have been linked to an increase in myomectomy complications over the past ten years [10].

Over the past decade, according to studies [3,11,12,13], two factors may have contributed to the rise in myomectomy complications over the past ten years: gynecologists performing MIS for myomectomy without adequate training in laparoscopic myomectomy techniques and myometrial suturing, and the rising incidence of fibroids in women around 40 years old seeking pregnancy, in which the uteruses are less and less elastic and more fibrotic for the phenomenon of uterine aging [14].

The most common intraoperative problems after laparoscopic myomectomy include excessive blood loss, myometrial hematoma, and morcellation mishaps. Insufficient use of vasoconstrictive agents, fibroid size, fibroid position, number of fibroids, failure to identify the feeding vessel, failure to identify the cleavage plane, inadequate hemostasis, loose knotting, inexperienced surgeons, and a lack of suturing precision are risk factors that contribute to intraoperative bleeding [10].

Problems resulting from insufficient myoma excision further enhance the likelihood of intermittent bleeding to menometrorrhagia, single episodes of discomfort, extremely severe dysuria, recurrent cystitis, constipation, bowel spasms, and continuous bleeding over a period of weeks. The significant bleeding and failure to control it are also the key grounds for conversion to laparotomy, which has been documented between 0.34% and 2.7% [15,16,17].

Myomectomies frequently present greater challenges and are associated with elevated intraoperative hemorrhage in comparison to hysterectomies, attributable to tumor-induced neovascularization and anatomical distortion. Factors such as the quantity, dimensions, and spatial distribution of the fibroids (Figure 2), in addition to the surgical approach employed, may significantly impact the likelihood of hemorrhage and the ensuing necessity for blood transfusion, ultimately leading to substantial blood loss during the procedure.

The worst risk during fibroid enucleation is the conversion of myomectomy into a hysterectomy, which would increase morbidity. Beyond blood transfusions, additional potential consequences of hemorrhagic complications during myomectomy encompass compromised hemostasis, circulatory shock, and mortality. Consequently, measures aimed at mitigating hemorrhage and its related complications throughout myomectomy are critically essential to reduce both morbidity and mortality [3,12,13,14,15,16,17].

Prior studies have reported variable rates of blood transfusions associated with abdominal myomectomy, ranging from 8.0% to 28.2%; however, these findings are based on procedures performed almost 20 years ago and are probably not indicative of current surgical procedures [18,19,20,21,22,23,24,25,26,27,28].

The risk of blood transfusions during myomectomy procedures is well established; however, the morbidity of perioperative transfusions following myomectomy procedures is not well understood. In any case, uterine hemorrhage following myomectomy has significantly decreased in recent years due to advancements in surgical equipment linked to innovative procedures and technology.

Kim T. et al. [29] analyzed 15,446 myomectomy surgeries in 2020 in order to determine the rate of blood transfusions during or within 72 h after myomectomy, evaluate risk variables related to getting a blood transfusion, and determine the surgical impact of blood loss following myomectomy. Compared to earlier reports in the recent literature, he found a greater rate of blood transfusions during or within 72 h following myomectomy. Furthermore, problems during the first 30 days following myomectomy were more common in women who got blood transfusions at the time of the procedure [29].

With the ongoing advancements in Assisted Reproductive Technology (ART), there is a likelihood that patient interest in preserving uterine integrity through alternative treatment modalities for symptomatic fibroids, rather than opting for definitive hysterectomy, may continue to escalate. Given that myomectomy is recognized as the benchmark for addressing symptomatic fibroids in women desiring fertility, it is imperative to implement strategic measures aimed at optimizing modifiable transfusion risk factors, which may mitigate the morbidity associated with perioperative blood transfusions. Authors have determined that the aggregate risk of requiring blood transfusion during myomectomy stands at 10%, and this is correlated with an elevation in postoperative morbidity within a 30-day timeframe [29]. This thorough analysis makes it clear how serious the problem of hemorrhage during myomectomy is; as a result, it is crucial to evaluate modern tools or techniques intended to reduce surgical bleeding during laparoscopic, robotic, or laparotomic myomectomy since controlling blood loss during this procedure continues to be a major challenge for reproductive surgeons. Over the years, numerous approaches have been outlined and promoted to achieve this goal.

## 4. The Biology of Fibroids’ Vasculature

Fibroids are predominantly reliant on the uterine arteries for the acquisition of their vascular supply; however, the uterus also benefits from supplementary blood flow provided by the ovarian arteries, which, in theory, could nourish a proximal fibroid. The presence of fibroids cultivates an extensive network of secondary vascular branches, which subsequently leads to the alteration and enlargement of both the caliber and trajectory of the uterine arteries [30,31]. Moreover, the literature highlights that alternative arteries, such as the omental artery, can supply fibroids in cases of uterine myomatosis. This underscores that ligation or clipping of the uterine artery alone may not always be sufficient or effective [30]. Similarly, the uterine arteries associated with fibroid-affected uteri exhibit augmented blood flow, diminished resistance index, and reduced pulsatility index when compared to the uterine arteries from non-fibroid-affected uteri [32].

Fibroids are characterized by a notable hypovascular nature at their centers and receive perfusion from diminutive peripheral arteries through arterial plexuses, with only a limited number demonstrating the presence of intra-fibroid arteries [33,34].

In clinical investigations, fibroids frequently appear in conjunction with systemic pathologies, particularly those that implicate endothelial or vascular dysfunction. This emergent correlation prompts intriguing foundational inquiries regarding potential causation and whether fibroids may serve as a uterine expression of a broader systemic disease process. Fibroids are fundamentally intertwined with vascular biology and may indeed contribute to vascular dysfunction [35].

These findings have substantial consequences for the long-term health prognosis of women diagnosed with fibroids and offer a new conceptual framework to commence the investigation of therapeutic methods aimed at fibroids. A selective layer of tissue between the fibroid and the surrounding myometrium, comprising the blood vessels necessary to sustain the fibroid mass, can form a highly vascularized pseudocapsule in response to fibroid growth despite the fibroid mass itself being poorly vascularized [36]. The fibroid is anchored to the pseudocapsule by connective bridges (Figure 3) but lacks its own true vascular pedicle.

Parallel arrays of bigger veins and exceptionally packed capillaries formed the pseudocapsule, which was isolated from the myometrial vasculature by a thin avascular gap, according to the inspection of the pseudocapsule and the surrounding myometrium (Figure 4) [37].

The veins surrounding the myoma circulate beneath the pseudocapsule, which is arranged in a plexus, and the biochemical growth factors evaluation in pseudocapsule vessels revealed intense angiogenesis [38]. The pseudocapsule vessels, which originate from the surrounding myometrium, throw themselves in a group to the center of the vascular network (Figure 5).

The development of a “protective” vascular capsule that supplies blood to the expanding tumor is most likely the result of the angiogenesis of the myoma pseudocapsule [39]. The myoma pseudocapsule, a type of neurovascular bundle that is abundant in neuropeptides and neurotransmitters, plays a critical role in innervation repair and wound healing, as well as in subsequent sexual and reproductive functions [40]. Neuropeptides and neurotransmitters, including substance P (SP), calcitonin gene-related peptide (CGRP), neuropeptide Y (NPY), oxytocin (OXT), vasopressin (VP), PGP 9.5, and growth hormone-releasing hormone (GHRH), are implicated in muscle repair phenomena and wound healing [41]. Since the majority of these neuropeptides have been highlighted in the myoma pseudocapsule, preserving them during myomectomy encourages appropriate myometrial post-surgical repair [42].

Transabdominal and transvaginal sonography reveals the pseudocapsule as an echogenic line encircling the myoma, with a wall that is at least one centimeter distinct and augmented by distal echoes. In ultrasound Doppler imaging, the myoma pseudocapsule manifests as a “ring of fire”, whereas histological images reveal a clear demarcation separating it from the myometrium. Although the myoma pseudocapsule’s vasculature may have structural flaws that make it brittle, the pseudocapsule is composed of the same cell types and exhibits the same biological structure as the nearby myometrium [37].

## 5. Traditional Techniques to Reduce Bleeding

Pharmacological options are typically required to control blood loss during surgery. These can be roughly separated into two categories: preoperative and intraoperative.

In an effort to reduce blood loss and shrink the fibroids, in the preoperative period, patients may be given aromatase inhibitors or gonadotropin-releasing hormone analogs prior to surgery [23].

Chemical and mechanical options are further subcategories of intraoperative possibilities. Mixed options include mechanical and pharmacological ones, such as intramyometral injections. These injection solutions serve a dual purpose during myomectomy: facilitating dissection by creating a clear surgical plane when injected into the subcapsular layer between the myoma and healthy myometrium and leveraging their chemical properties to achieve vasoconstriction (as with vasopressin) or to minimize blood loss through alternative mechanisms (such as oxytocin and tranexamic acid).

## 6. The Chemical Options to Reduce the Bleeding During Myomectomy

Chemical options include injecting a normal saline solution, administering drugs (oxytocin, tranexamic acid), or injecting a vasoconstrictive agent (vasopressin) into the myoma’s intracapsular layer. There is currently low-quality evidence that bupivacaine plus epinephrine, tranexamic acid, gelatin-thrombin matrix, ascorbic acid, dinoprostone, loop ligation, fibrin sealant patches, peri-cervical tourniquet, or tourniquet tied around both the cervix and infundibulopelvic ligaments may reduce bleeding during myomectomy, and moderate-quality evidence that misoprostol or vasopressin may lessen bleeding during myomectomy [43]. Moreover, vasopressin is prohibited in several nations due to the possibility of fatal cardiovascular collapse. The GnRH analogs, which have been used for many years to lessen the size of fibroids and uterine hemorrhage before surgery, are costly and linked to menopausal adverse effects after extended use. They have been demonstrated to have no effect on the volume of blood lost after surgery despite the fact that they shrink the size of the fibroid masses. Due to the possible loss of the operative plane and the resulting increased difficulty in performing the surgery, the efficacy of this has been questioned since they cause the planes of the myoma pseudocapsule to blur, which makes it harder to shell the mass. Selective progesterone receptor modulators are among the more recent drugs being studied. The smaller masses also shrink prior to surgery, but they reappear when the treatment is stopped [44].

There is no proof that morcellation, oxytocin, or temporary uterine artery clipping stops blood loss [43]. Now, we will discuss the most common chemical options to reduce bleeding during myomectomy.

### 6.1. Vasopressin

Vasopressin, a synthetic derivative of an anti-diuretic hormone that has been used for more than 70 years [45,46], has been studied in many studies to lessen blood loss during myomectomy [47] because it acts as a hormonal tourniquet. In fact, injecting intramyometral vasopressin at the site of the planned uterine incision for each fibroid reduces blood loss [48]. Dillon was the first to use vasopressin in 1962 [45] and discovered that 72% of patients who needed myomectomy did not need blood replacement when compared to control subjects. Frederick et al. reported much less blood loss than the untreated group [45].

An intra-myometrial vasopressin injection may successfully lower intraoperative blood loss, the need for blood transfusions, and the postoperative decline in hemoglobin and hematocrit during a myomectomy procedure [49]. Additionally, it removes the possibility of unintended, irreversible uterine ischemia damage and thrombo-embolic events from mechanical procedures [50].

Additionally, the vasoconstrictive effects of vasopressin begin to wear off if fibroid dissection from the myometrium is not completed within the 20 min given, which may result in greater bleeding. Furthermore, vasopressin has no effect on large blood vessels. A potent systemic vasoconstrictor, vasopressin can cause hypertension and bradycardia. To avoid unwanted blood pressure rises, hemodynamic monitoring should be employed when infiltrating a diluted fluid. Vasopressin has an anti-diuretic effect that lasts for two to eight hours [45,46]. A number of European nations, including France and Italy, have removed vasopressin from the pharmaceutical market due to safety concerns. Vasopressin administration is technically a straightforward and safe process. Due to its affordability and accessibility, vasopressin can be utilized often during laparoscopic or abdominal myomectomy [46,48].

In order to gather data from randomized controlled trials (RCTs) and nonrandomized controlled trials (NCTs) regarding the safety and effectiveness of vasopressin versus passive control (placebo/no treatment) during myomectomy, Alomar et al. [47] examined eleven studies in 2022, involving 1067 patients (vasopressin = 567 and control = 500). When compared to a passive control intervention, the prophylactic injection of vasopressin was generally safe and associated with considerable reductions in intraoperative blood loss and concomitant morbidities among patients undergoing myomectomy. However, the conclusions should be regarded cautiously due to the low quality of the data and the significant variance in the doses used among studies. Additionally, it is occasionally linked to intraoperative vasoconstriction and hypertension as a result of systemic absorption [51,52,53].

### 6.2. Octreotide Acetate

Octreotide Acetate (OA) is an octapeptide that pharmacologically resembles natural somatostatin, and it is frequently used as an adjuvant to treat variceal hemorrhage [54]. Human investigations conducted in vivo have demonstrated that OA directly affects the smooth muscle of the arteries and veins, causing vasoconstriction [55]. Additionally, it was determined that in individuals with cirrhosis, OA has a local arterial vasoconstrictive impact that is not dependent on systemic hormonal control of glucagon; this vasomotor activity appears to be independent of nitric oxide [56].

A study was conducted to assess the efficacy of local injection of OA in reducing blood loss during conventional abdominal myomectomy and to compare it with local instillation of vasopressin [57]. Three groups of 20 cases each were randomly selected from 60 cases with symptomatic leiomyomata who were scheduled for abdominal myomectomy. Group I received a local vasopressin injection; Group II received OA; and Group III (the control group) received no medicine at all and just received a local injection of free saline. According to the overall findings, the OA group experienced a much shorter operating time and less blood loss than the control group. However, operational duration and blood loss were significantly higher than in group I instances. Group III had significantly lower postoperative hemoglobin levels than groups I (*p* = 0.039) and II (*p* = 0.044). None of the cases in group III required a blood transfusion, compared to just one in group II and none in group I. Despite its potential to reduce blood loss following myomectomy, local intra-myometrial OA was not as effective as local vasopressin. The low doses and decreased vasoconstrictor action of OA may help to explain this [57].

### 6.3. Tranexamic Acid

In a variety of surgical operations, including orthopedic, cardiac, general, gynecologic, and obstetric procedures, as well as organ transplant surgeries, tranexamic acid (TXA) has gained popularity for minimizing blood loss [58,59].

TXA has been clinically utilized extensively in gynecology to prevent severe menstrual bleeding [24,60]. Tranexamic acid decreased the risk of blood transfusion by 34% without increasing the risk of venous thromboembolism (VTE), a known side effect of the medication, or other unfavorable perioperative outcomes, according to systematic reviews of randomized control trials (RCTs) involving more than 25,000 patients who used the medication for elective surgery. This is most likely because TXA acid prevents the enzymatic breakdown of fibrins in blood clots by blocking the conversion of plasminogen to plasmin, which lowers blood loss [61].

Olaleye et al. [62] assessed the efficacy of tranexamic acid in reducing myomectomy-associated blood loss by a prospective double-blinded randomized trial conducted on women who had abdominal myomectomy. Tranexamic acid (TXA) was administered intravenously to the study group during surgery, and a placebo was given to the control group. Weighing the surgical swabs and monitoring the suction apparatus’s volume allowed us to compute the intraoperative blood loss. Blood drawn from the drapes and wound drains after surgery was also measured. For every case, hemoglobin levels were measured both before surgery and on the second postoperative day. No negative effects were observed in either group. Since TXA lowers the overall blood loss associated with myomectomy, improves postoperative hemoglobin concentration, and lowers the risk of blood transfusion requirements, this study has shown the effectiveness and safety of perioperative 1000 mg TXA in reducing blood loss during open myomectomy. This is likely because TXA inhibits the enzymatic breakdown of fibrins in blood clots by blocking the activation of plasminogen to plasmin. Additionally, it is linked to shorter hospital stays. Thus, prophylactic TXA was therefore stated as a safe and efficient way to minimize blood loss during open myomectomy [62].

### 6.4. Uterotonics

The use of uterotonics like misoprostol, ergometrine and oxytocin have been used during myomectomy to prevent excessive blood loss, even if we have already mentioned that a non-pregnant uterus has a significantly smaller number of oxytocin receptors than a pregnant one [63,64]. In fact, a non-pregnant uterus has a significantly smaller number of oxytocin receptors than a pregnant one. Although there are many oxytocin receptors in a pregnant uterus, in a non-pregnant uterus, which is the usual situation when conducting a myomectomy, their concentration is 50–100 times lower [64].

Matasariu et al. [65] performed an observational study to compare cases of laparoscopic myomectomy performed with and without the use of glypressin, a synthetic form of Vasopressin, or any other vasoconstrictor drug in order to determine if the use of this medication played a significant influence in myomectomy. Glypressin 0.2 mg/mL was injected intramyometrically into one group of women (n = 64) in a volume of 5 mL (glypressin group; group 1). In order to minimize blood loss, a laparoscopic myomectomy was performed on the other group of women (n = 124) (group 2). Blood loss and the postoperative decrease in Hb and Htc were significantly reduced with the administration of intramyometral glypressin. The authors concluded that glypressin works well to lessen blood loss during laparoscopic myomectomy. Since blood pressure and pulse rate were within the same normal ranges as in the control group, there were no adverse effects of glypressin administration on the circulatory system in this observational investigation [65].

## 7. The Mechanical Options to Reduce the Bleeding During Myomectomy

One of the earliest methods to reduce blood loss was to have a colleague hold the broad ligaments tightly with each hand to stop blood flow via the uterine veins. Victor Bonney first used a specifically made clamp in the 1920s [66], and Rubin was the first to apply an elastic rubber tourniquet in 1938 [66].

Liu et al. initially reported surgically blocking the uterine arteries to treat symptomatic uterine fibroids in 2001 [67].

In the context of myomectomy for the treatment of uterine fibroids, mechanical alternatives include temporary uterine artery clipping, tourniquet implantation, commonly by a Foley catheter or a Penrose drain, and embolization of the uterine artery (UAE).

These techniques have pros and cons, and the decision between them is influenced by a number of variables, including the patient’s health, the location and size of the fibroids, and the surgeon’s level of experience [68].

Two recent meta-analyses found that uterine artery closure significantly reduces blood loss and transfusion [69], whereas a 2014 Cochrane review [43] on the subject found that temporary uterine artery cutting does not reduce blood loss.

### 7.1. Uterine Artery Clamp

There is a great deal of variation in the uterine vascularization. Because it receives its blood supply from a variety of sources, the uterine vascular network is extensive. The myometrium has a circulatory supply that is incredibly thick. First, the uterine arteries, which have sizes of roughly 2 to 6 mm, are primarily responsible for delivering blood to the uterus [70].

Subsequently, diminutive communicating arteries, measuring about 0.5 mm, establish connections between the uterus and the ovarian arteries [71].

Furthermore, several named arteries possess the capability to provide blood to the uterus, including the inferior mesenteric, lumbar, vertebral, middle sacral, deep iliac circumflex, inferior epigastric, medial femoral circumflex, and lateral femoral circumflex arteries [72]. Finally, a multitude of extremely small, unnamed arteries supply blood to the uterus, originating from the broad ligament and the retroperitoneal space [73].

The uterine arteries are the only source of vascular supply to the fibroids, in contrast to the uterus, which has several blood sources. Since the uterine arteries allow the majority of blood to enter the uterus, it was hypothesized that transitory uterine ischemia would occur if the arteries were blocked via a catheter or a laparoscopic procedure [74].

By controlling the blood flow to the fibroids, artery uterine clipping helps the surgeon minimize bleeding during the myomectomy [75,76].

Additionally, artery uterine clipping typically takes more ability than the typical reproductive surgeon to dissect, identify, and clamp the uterine arteries [77].

No research has been conducted on the long-term effects of permanent occlusion on those who want to become pregnant in the future. Because permanent uterine artery occlusion during surgery is comparable to uterine artery embolization, caution must be used due to the unknown possible negative effect on fertility and obstetric outcomes associated with uterine artery embolization [78].

To clarify the function of prophylactic uterine artery occlusion during a laparoscopic myomectomy, Dubuisson et al. [79] reviewed the literature. According to their findings, patients who had preventive uterine artery blockage during surgery experienced a significant reduction in operative blood loss in six out of eight comparable investigations. Temporary clipping reduces the chance of tissue injury by lowering the danger of ischemia. However, the surgeon must go through a learning curve for operating skills, including accessing the retroperitoneum and recognizing the pelvic vasculature in order to use these vascular clamping techniques with vascular clips [79].

Sanders et al. [68] conducted a systematic review and meta-analysis that included randomized and observational trials to ascertain the efficacy of concurrent surgical uterine artery occlusion in women undergoing myomectomy because the decrease in blood loss following uterine artery occlusion at myomectomy has not been consistently demonstrated.

Measures of intraoperative blood loss, including transfusion rate, change in laboratory hemoglobin levels, and predicted blood loss, are the main results. Both short-term (operative duration, length of hospital stays, and complication rates) and long-term (symptom change and fibroid recurrence) secondary surgical outcomes were assessed in twenty-six studies involving 2871 patients. When compared to controls, the authors discovered that surgical uterine artery occlusion, during laparoscopy or laparotomy, at the time of myomectomy statistically significantly reduced estimated blood loss (mean difference 103.7 mL, *p* < 0.001), blood transfusion risk (RR 0.24, *p* < 0.001), and postoperative hemoglobin change (mean difference ~0.60 g/dL, *p* < 0.001). A shorter hospital stay and a lower risk of fibroid recurrence when compared to controls were two additional benefits of uterine artery closure. On average, myomectomies with uterine artery occlusion required 10.9 more minutes of surgery. The risk of surgical complications was not increased by the use of uterine artery blockage.

The lateral, posterior, and anterior techniques are the three primary methods used to conduct uterine artery occlusion. Following surgical uterine artery occlusion, there is a decrease in arterial pulse pressure and uterine perfusion, which aids in hemostasis and reduces blood loss. Uterine ischemia is avoided by the pelvis’s extensive collateral circulation. A competent surgeon can do this surgery in 10 to 20 min, and it is also reasonably priced and suitable for settings with limited resources. However, it’s crucial to weigh the benefits of uterine artery occlusion against the risks, which primarily rely on the surgeon’s ability and knowledge [68].

### 7.2. Tourniquet

In the 1920s, Victor Bonney introduced a specially designed clamp and Rubin, in 1938, was the first to use an elastic rubber tourniquet [66]. One tourniquet is placed around the cervix in the pericervical region to compress the uterine isthmus, which narrows the uterine arteries. During a laparoscopic or laparotomic myomectomy, applying a tourniquet is an easy process that does not require specialized equipment or highly qualified surgeons. Reducing bleeding allows the surgeon to see the surgical site more clearly. The tourniquet is by far the least expensive and eco-friendly way to stop blood loss during surgery. For this purpose, a Penrose drain or the conventional Foley’s urethral catheter can be easily modified [80,81].

Additionally, there was no proof that the uterine tourniquet had a detrimental effect on the pregnancy’s outcome. Applying a tourniquet during abdominal myomectomy significantly reduced the mean quantity of blood loss when compared to the non-tourniquet group, but it did not reduce the ovarian reserve or shorten the surgical time. It is safe and efficient to apply a tourniquet during an abdominal myomectomy [82].

Abdul et al. [83] performed a randomized, double-blind controlled investigation to ascertain whether using perioperative intravenous TXA in addition to tourniquets can further reduce blood loss during abdominal myomectomy. Using basic random selection, participants were randomized to either the tourniquet plus intravenous TXA or the tourniquet plus placebo groups. Intraoperative blood loss, postoperative hemoglobin levels, and the requirement for intraoperative blood transfusions were the main results. When compared to the tourniquet alone, the results demonstrated that the addition of TXA to the tourniquet considerably lowers intraoperative blood loss during abdominal myomectomy [83]. TXA was advised as an adjuvant to further minimize intraoperative blood loss during abdominal myomectomy, as recently reported by some Nigerian authors [84].

### 7.3. Uterine Artery Embolization

Compared to a standard myomectomy, uterine artery embolization (UAE) requires no significant surgery, meaning shorter hospital stays and quicker recovery periods. There is very little blood loss during the operation because it is a non-surgical method. However, big fibroids or those deep within the uterine wall might not respond as well to a UAE [85].

Research assessing the effects of UAE has revealed a higher chance of miscarriage and a decrease in conception rates [86].

There are pathophysiological repercussions to the decrease in uterine vascular perfusion brought on by the blockage of the uterine arteries [87]. The long-term reproductive consequences of reduced uterine perfusion have been linked to the hypothesis that uterine artery embolization or occlusion results in tissue hypoxia and ischemia, which triggers cellular death of remaining fibroids [88].

## 8. A Biology-Based Technique to Reduce Myometrial Bleeding During Myomectomy: The Intracapsular Myomectomy

The most often overlooked factor is proper technique, as surgeons may unintentionally dissect in the wrong anatomical plane, which could result in significant bleeding. The appearance of the innermost layers of the myometrium, which may appear unduly pale when stretched over the myoma, can be attributed to the inability to detect this problem. In addition to minimizing blood loss, efficient hemorrhage control during myomectomy facilitates access to the precise plane of dissection, which lowers the need for energy sources, avoids tissue necrosis, and improves postoperative healing. In order to reach the proper anatomical plane to remove fibroids from the myometrium, the inexperienced surgeon mistakenly believes they are inside the pseudocapsule and begins myoma enucleation while several more layers need to be passed. Since all of the techniques discussed thus far have focused on surgical and pharmaceutical practices in general rather than the unique characteristics of the uterus and its ultrastructure, the biology and anatomy of the uterus must be the first step in determining the best method to reduce myometrial bleeding during myomectomy.

Uterine fibroids, during their growth, facilitate the progressive development of a peripheral surgical-anatomical biological structure known as myoma pseudocapsule [37,38]. This anatomical entity, which remained largely unrecognized for numerous years, has garnered significant interest among gynecologists specializing in fertility surgery in the contemporary century. This biological structure arises from the compression of the surrounding myometrium by the fibroid, creating a separation, albeit minimal, between the fibroid and the healthy myometrium. In succinct terms, the pseudocapsule displaces the intact myometrial muscular fibers, functioning as a delicate fibromuscular barrier, thereby preserving the integrity and contractility of the uterine musculature. Moreover, from an anatomical perspective, the uterine fibroid is structurally linked to its pseudocapsule via connective bridges; however, it is devoid of its own true vascular pedicle and, on occasion, possesses vascular and collagen fiber bridges that anchor the myoma to the myometrium, thus disrupting the surface of the pseudocapsule [37] (Figure 6).

This arrangement culminates in the establishment of a distinct cleavage plane that exists between the myoma and the pseudocapsule, commonly referred to as the cleavage plane. Biologically speaking, the myoma pseudocapsule exhibits a bio-structural composition analogous to that of the myometrium.

At the ultrastructural level, as observed through transmission electron microscopy (TEM), the cells within the pseudocapsule exhibit characteristics akin to smooth muscle cells, akin to those of the myometrium, thus suggesting that the pseudocapsule constitutes a segment of the myometrium compressed by the myoma [42].

The most noteworthy biological activity observed in the myoma pseudocapsule pertains to its neoangiogenic capabilities. Indeed, the myoma pseudocapsule is abundant in collagen fibers, neurofibers, and blood vessels, akin to a neurovascular bundle encasing the fibroid. Within this neurovascular bundle, several neoangiogenic factors have also been investigated. Angiogenetic factors detected in the vessels of the pseudocapsule are already extensively implicated in the physiological functions of the myometrium [39].

The evaluations of these biochemical growth factors have indicated pronounced angiogenesis within the vessels of the pseudocapsule, and these substances are postulated to play a crucial role in wound healing and muscular innervation. Consequently, the growth factors present in the myoma pseudocapsule are thought to instigate peripheral angiogenesis relative to the myometrium.

Therefore, we proceeded to examine a critical phase of uterine physiology, namely wound healing, which is also predicated upon muscular innervation. Myometrial wound healing is characterized as an interactive and dynamic process that relies on neuromodulators, angiogenetic factors, neuropeptides, blood cells, extracellular matrix components, and parenchymal cells, manifesting through three intricate and overlapping phases: inflammation, tissue formation, and tissue remodeling [41]. Within the physiology of these processes, the nervous system and its neurotransmitters are also integrally involved [41]. These substances play a significant role in mediating inflammation and wound healing, contributing to physiological processes and scar repair across various tissues, including the uterine muscle. In regenerative processes associated with the preservation of the pseudocapsule, such as in the context of post-myomectomy uterine scarring, neuropeptides and neurotransmitters are speculatively implicated in the processes of wound healing and myometrial regeneration [41].

Thus, the intracapsular myomectomy procedure is predicated upon the principles of muscular physiology while concurrently adhering to anatomical considerations. By selectively coagulating the fibroneurovascular bundles of the network surrounding the fibroid, the proper application of the surgical method to biology invariably reduces blood loss during intracapsular myomectomy. Following the incision of the uterine serous surface, one continues deep until they reach the fibroid pseudocapsule, which coagulates and is cut to hook the fibroid and gently drag it outwards while coagulating and dissecting the pseudocapsule’s fibroid neurovascular bundles (Figure 7).

Due to the fibroid being vascularized solely through the pseudocapsule’s veins, there can be very little bleeding before and after surgery. The objective remains the selective coagulation of the fibroneurovascular bundles during myomectomy, which allows for the reduction of bleeding to a minimum.

The proper intracapsular myomectomy approach, which selectively coagulates the pseudocapsule vessels, is undoubtedly responsible for the decreased intraoperative blood loss rather than the use of vasopressin or uterine artery occlusion [89].

Selectively coagulating the vessels that vascularize the fibroid during enucleation is the issue, not decreasing vascular flow to the fibroid and changing myometrial physiology. Additionally, CO_2_ aids the surgeon during myomectomy by enabling a procoagulant effect during laparoscopy [89]. Furthermore, the only difference between a single and many myomectomies is the length of time the intervention is executed (the multiple myomectomies take longer).

The intracapsular myomectomy may be executed through various approaches, including laparotomy (Figure 8), laparoscopy, robotic assistance, vaginal access (Figure 9), and hysteroscopy.

Furthermore, this surgical intervention can be performed safely during a cesarean section, referred to as cesarean myomectomy [90].

The advantages of this surgical technique are apparent both intraoperatively and postoperatively: there is a notable reduction in hemorrhage, the integrity of myometrial anatomy is largely preserved, and the healing processes of the myometrium are maintained and potentially enhanced, as substantiated by clinical assessments and ultrasound evaluations of the scar site subsequent to intracapsular myomectomy [41,91].

Unfortunately, another common misconception is that there is a vascular pedicle at the fibroid’s base. This misinterpretation is just as harmful as not recognizing the proper dissection plane. After performing dissection—often in an incorrect plane—in a blood field, the surgeon quickly tries to block the vascular pedicle at the base of the myoma, ignoring the continuous bleeding from the myometrial defect that has been formed. The vascular supply to the myoma is circumferential and unpredictable, characterized by vessels of variable caliber that provide nourishment and drainage. Recognizing and meticulously dissecting in the appropriate plane beneath the pseudocapsule, in conjunction with securing the blood vessels as they are encountered during dissection, constitutes the most pivotal measures to minimize blood loss during a myomectomy [92].

The surgical protocol for laparoscopic/robotic or laparotomic intracapsular myomectomy is summarized as follows. These are the main surgical concepts: (1) the exclusion of preoperative treatments to prevent alterations in the fibroid’s texture and structure, as well as fragmentation of the pseudocapsule; (2) the intracapsular myomectomy technique is conducted in accordance with the mentioned literature; (3) the operation is performed without the application of chemical or mechanical traditional options to reduce the bleeding during myomectomy (as vasopressin or a tourniquet, or uterine artery clamps, tranexamic acid or misoprostol); (4) an adhesion barrier should be utilized after hysterorraphy. The intracapsular myomectomy can be executed in all patients without limitations, with a consistent emphasis on identifying the cleavage planes of the fibroids, thereby enucleating the fibroids from their pseudocapsule and ensuring minimal trauma to the myometrium. The visceral peritoneum is incised along the midline longitudinal plane utilizing monopolar scissors, an ultrasonic scalpel, or a monopolar crochet needle electrode. The plane for myoma enucleation is delineated and dissected using monopolar scissors, crochet needle, ultrasonic scalpel, or bipolar clamps, while the assistant employed an irrigator cannula to enhance the panoramic laparoscopic visibility of the pseudocapsule surrounding all subserous-intramural leiomyomas.

Upon visualization of the free myoma surface, it is grasped using a Collins laparoscopic forceps or myoma screw to facilitate the necessary traction for its enucleation from the pseudocapsule. Hemostasis of the small vessels is delicately achieved utilizing a bipolar clamp, ultrasonic scalpel, monopolar scissors, or monopolar crochet, aiming to liberate the connective bridges of the fibroid from its pseudocapsule. Generally, the thorough enucleation of the fibroid from the pseudocapsule is accomplished with minimal blood loss.

In instances involving pedunculated fibroids, the pedicle was coagulated using bipolar forceps at a distance of 1–2 cm from the uterus towards the fibroma, enabling selective coagulation and transection of the vessels with an ultrasonic scalpel, scissors, or hook, occasionally employing loops or staples, with suturing typically not necessitated. In cases of deep intramural myomas, chromopertubation may be employed via a cervical cannula, not solely to assess tubal patency but also to aid in the direct identification of any inadvertently opened uterine cavity.

The closure of the myometrium may be executed through various methodologies. Traditionally, this can be achieved via a single or double layer utilizing 1/0 braided or monofilament thread, with a round CT-1 carved needle, deploying either intra- or extracorporeal knots. In sub-serosal myomectomies, the edges of the uterine defect are approximated utilizing introflexion single U-stitches at 1 cm intervals from the incision edge, following a baseball-type suture technique. Deep intramural fibroids necessitated a two-layer closure of the myometrium employing introflection sutures. Should the uterine cavity be opened during the fibroid enucleation, 2–3 deep myometrial single or continuous sutures are applied. The overlying serosa is repaired using multiple single or continuous introflexion stitches. Alternatively, the myometrium may be closed with a single layer utilizing unidirectional running barbed sutures. The edges of the superficial uterine defect were approximated with introflexion U-inverted stitches (myometrium/serosa-serosa/myometrium orientation) at 1 cm intervals (a baseball-type suture). In laparotomy, myography can be conducted in a single layer utilizing 1/0 monofilament (poliglecaprone 25), which may include the overlying serosa or be performed multilayered when deemed appropriate.

## 9. Discussion

Some basic principles must be remembered in the present surgical practice of myomectomy in order to properly handle the procedure. It is not always possible to use vasopressin in every therapeutic situation. Vasopressin’s main disadvantage when used during surgery is that its pharmacological mechanism causes brief periods of bradycardia and hypertension, which, if left untreated, could cause significant morbidity. Case reports of pulmonary edema, atrioventricular obstructions, and cardiac arrest following vasopressin administration are available. Empirical research has indicated that cardiovascular complications may be mitigated by refraining from the use of vasopressin in individuals with pre-existing heart conditions, appropriately diluting the dosage with saline, preventing inadvertent intravascular injection of vasopressin, and meticulously monitoring vital signs during administration [48].

Therefore, in situations when vasopressin is contraindicated, the surgeon must have other strategies to reduce intraoperative bleeding and facilitate myomectomy. Preventive uterine artery clamp or occlusion myomectomies have been shown to be effective in reducing intraoperative and postoperative hemorrhage. Bipolar coagulation and vascular clamps are two permanent ways to block the uterine arteries. As an alternative, clamps, detachable clips, or silk ties can be used to create temporary occlusion. Research tends to attribute the observed reduction in surgical blood loss to techniques that effectuate permanent occlusion of the uterine arteries during myomectomy. Furthermore, permanent occlusion has been correlated with a decreased likelihood of myoma recurrence; however, it is essential to acknowledge that this may pose negative implications for subsequent pregnancies [79].

The ongoing discussion about the potential effects of uterine devascularization on future reproductive outcomes is a major concern regarding uterine artery clamp or occlusion after laparoscopic myomectomy. Since the obstruction of the uterine arteries is thought to be the cause of only a 40% decrease in uterine vascularization, the dominant theory proposes that collateral vascular networks and anastomoses will restore and enhance healthy uterine tissue [92]. Nonetheless, further investigation is warranted to elucidate the effects on ovarian reserve, the trajectory of pregnancy, and potential delivery complications. These apprehensions are comparatively less pronounced in procedures that involve transient uterine artery clamps or occlusions during laparoscopic myomectomies. Both loop ligation and uterine artery clamp or occlusion techniques have been associated with enhanced quality of uterine suturing attributable to diminished intraoperative blood loss [89].

There is a burgeoning interest in the preoperative hormonal management of myomas, predominantly utilizing drugs as gonadotropin-releasing hormone (GnRH) analogs or selective progesterone receptor modulators such as ulipristal acetate (UPA).

Generally, GnRH analogs and selective progesterone receptor modulators are known to lower serum concentrations of estrogen and progesterone, respectively. Consequently, both hormonal interventions have been demonstrated to alleviate abnormal uterine bleeding associated with the presence of fibroids, rectify preoperative anemia, reduce fibroid volume, and diminish intraoperative hemorrhage [93].

Although GnRH analogs are correlated with a more pronounced reduction in fibroid size, one notable disadvantage is their potential to induce severe estrogen withdrawal and menopausal symptoms, including hot flashes, sleep disturbances, vaginal dryness, and irritability. Patients may exhibit diminished adherence to this therapeutic regimen due to its adverse effects. Although UPA is recognized for a comparatively milder side-effect profile, a salient concern is the occurrence of reported instances of liver toxicity associated with its use [79].

Another area of concern regarding UPA and GnRH analogs is their ability to alter fibroid structural integrity in a manner that complicates traction maneuvers during myomectomy, as well as diminishes the demarcation between the fibroids and the surrounding tissues [44,93]. Furthermore, both hormonal therapies have been linked to substantial financial burdens. Regarding the limitations of the study, the generalization of the proposal of intracapsular myomectomy as the “gold standard” requires additional validation through prospective multicenter studies. Additional research is necessary to evaluate the effectiveness of preoperative hormonal interventions relative to alternative methodologies employed in laparoscopic myomectomy procedures.

## 10. Conclusions

To mitigate the frequency of myoma recurrence and associated complications such as hemorrhage, hematoma formation, adhesions, and perforation of the gravid uterus, it is imperative to prioritize both specialized training and clinical experience. The surgical methodology employed for myomectomy is fundamentally dependent on the imaging diagnostics that provide insights into the fibroids and their pertinent anatomical features. The appropriate length of the hysterotomy in relation to the size of the myoma should be judiciously determined and should not be excessive, as it is essential to consider the myometrial elasticity (the greater the extent of muscle incision, the higher the likelihood of hemorrhage), which can effectively diminish bleeding, coagulation duration, and suturing times. We demonstrated, in this narrative review, that the enucleation of the fibroid can invariably be executed utilizing the intracapsular technique, thereby circumventing the necessity for traditional chemical and mechanical methods to prevent bleeding. During the enucleation process, all vascular structures are selectively coagulated, and the fibrovascular bridges are meticulously incised. There exists no requirement for the application of conventional mechanical or pharmacological hemostatic interventions during intracapsular myomectomy, as the technique itself inherently safeguards the myometrium and minimizes hemorrhage throughout the myomectomy procedure. The selective bipolar coagulation of the vessels within the pseudocapsule, while meticulously avoiding carbonization and ensuring the integrity of the myometrium post-suturing, may significantly reduce the likelihood of myometrial hematoma formation. The suturing of the myometrium must be executed with caution to prevent excessive traction on the muscle, thereby averting tissue ischemia, which could lead to subsequent hypoxia and ultimately result in tissue necrosis.

## Figures and Tables

**Figure 1 medsci-13-00068-f001:**
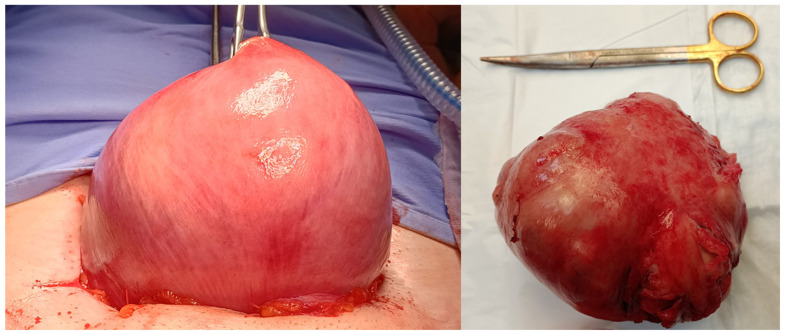
Image of traditional laparotomic myomectomy on the left and removed fibroid on the right.

**Figure 2 medsci-13-00068-f002:**
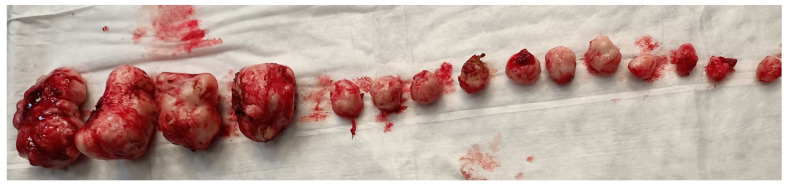
Removal of multiple fibroids may result in significant blood loss and the need for blood transfusions during surgery and in the postoperative period. Removal of multiple fibroids may result in significant blood loss and the need for blood transfusions during surgery and in the postoperative period.

**Figure 3 medsci-13-00068-f003:**
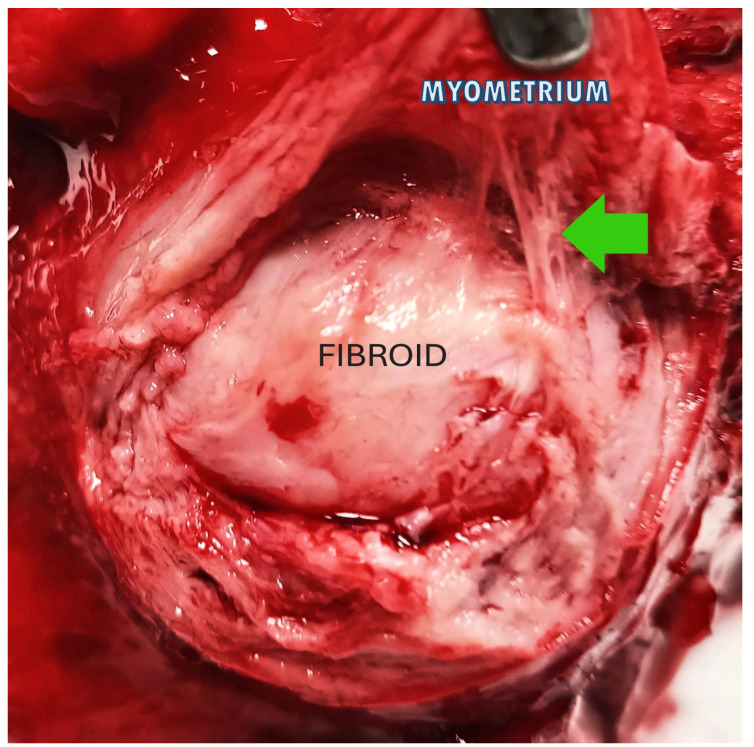
The image shows a fibroid inside the uterus: the green arrow highlights the connective bridges that anchor the fibroid (in the center) to the pseudocapsule, above which the myometrium appears.

**Figure 4 medsci-13-00068-f004:**
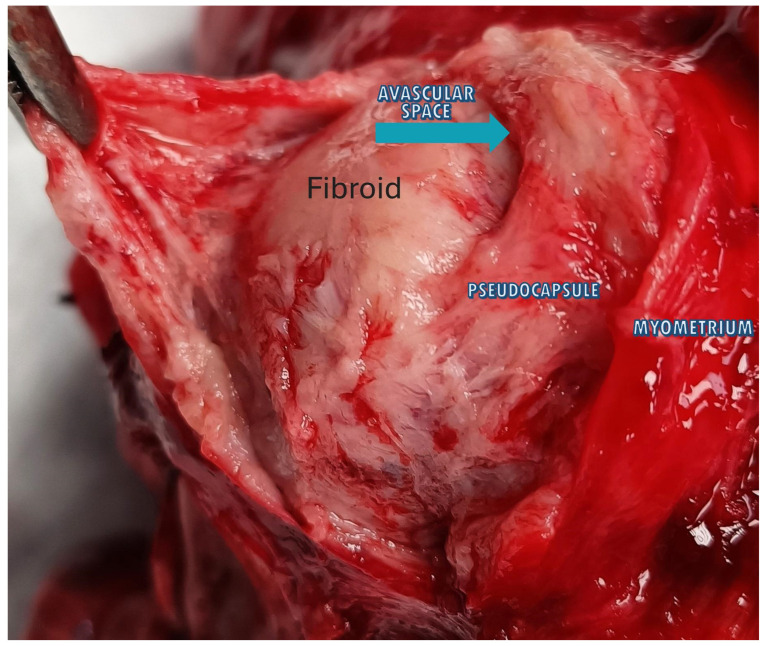
The image shows a fibroid inside the uterus with the 3 anatomical planes: fibroid, pseudocapsule and myometrium. The blue arrow highlights the avascular space that divides the fibroid from the pseudocapsule, above which appears the myometrium.

**Figure 5 medsci-13-00068-f005:**
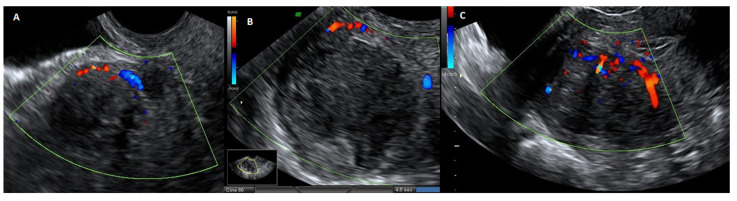
Ultrasound image with echodoppler of fibroma with overlying pseudocapsule: (**A**) on the left, vascular ring of fire of the pseudocapsule, which is included in the vascular ring; (**B**) in the center, fibroma without internal vascularization; (**C**) on the right, the vascular ring of fire of the pseudocapsule, with vessels detaching from the pseudocapsule and penetrating from the periphery of myoma towards the center of the fibroma.

**Figure 6 medsci-13-00068-f006:**
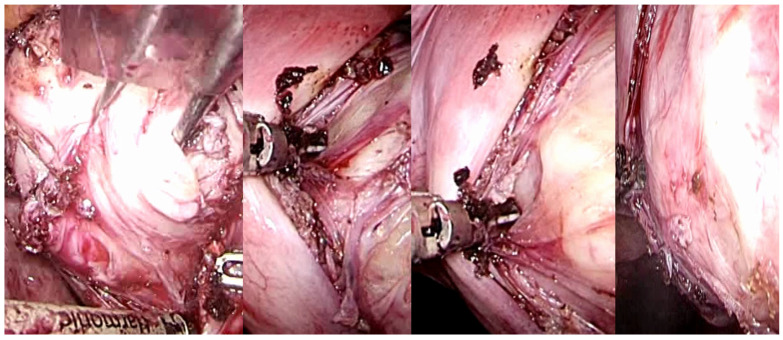
Laparoscopic image of intracapsular myomectomy. From left, progressive identification of the fibroid (in white, hooked with Collins’s forceps) with the pseudocapsule branches covering it and subsequent incision of the fibrovascular branches of the pseudocapsule to detach and enucleate the fibroid from the pseudocapsule, with minimal blood loss and less myometrial damage.

**Figure 7 medsci-13-00068-f007:**
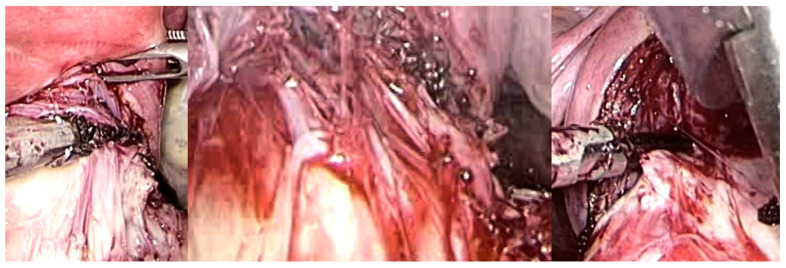
Laparoscopic image of intracapsular myomectomy. From left, progressive identification of myoma pseudocapsule branches covering fibroid and their subsequent incision to detach and enucleate the fibroid from the pseudocapsule, with better preservation of the myometrium and neurovascular bundle.

**Figure 8 medsci-13-00068-f008:**
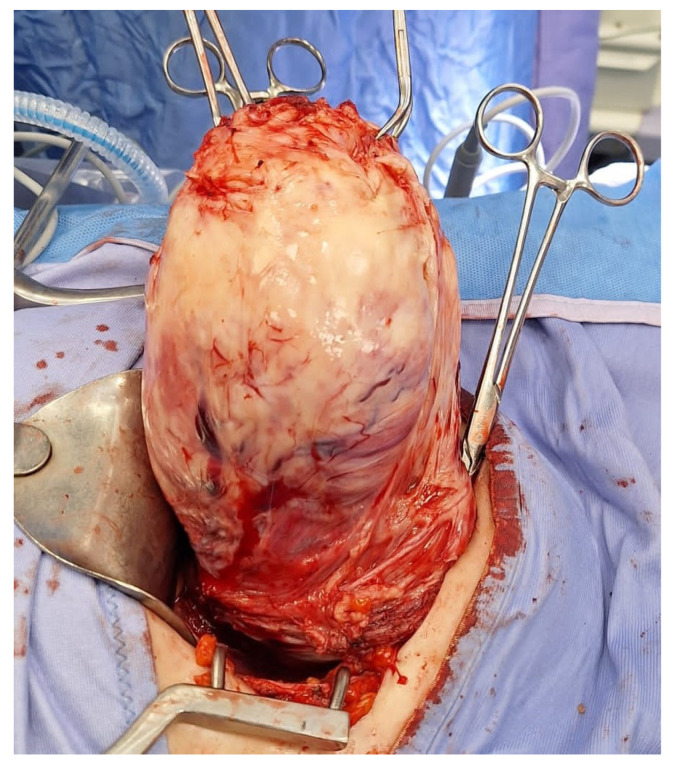
Laparotomic image of intracapsular myomectomy. The surgeon enucleates a large fibroid measuring over 15 cm, gradually detaching it from the uterus by sectioning bundles of pseudocapsule that anchor it to the uterus. Minimal bleeding is noted during myomectomy.

**Figure 9 medsci-13-00068-f009:**
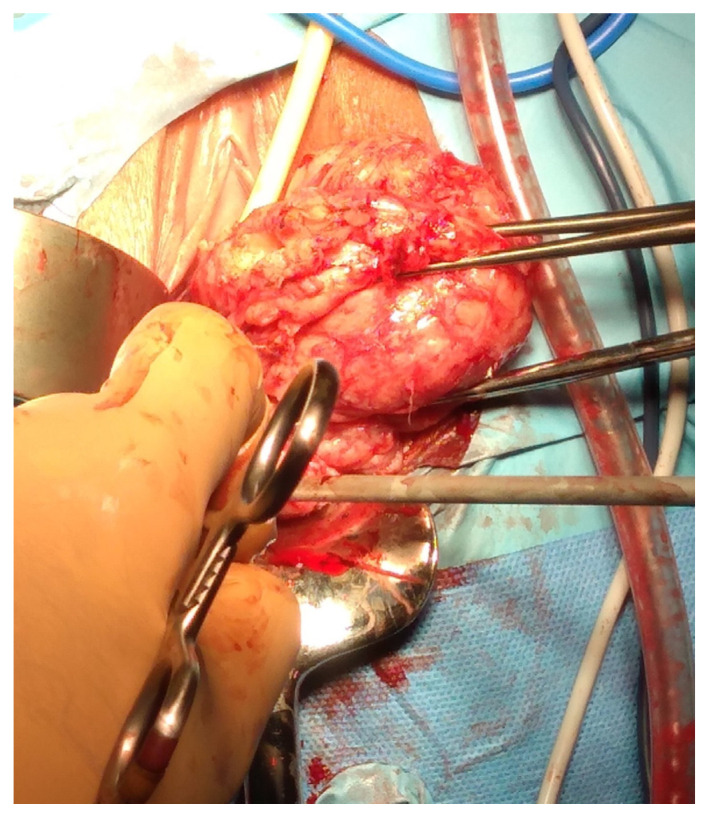
Vaginal intracapsular myomectomy. The surgeon enucleates from vaginal access a cervical fibroid measuring 8 cm, gradually detaching it from the uterus by sectioning bundles of pseudocapsule that anchor it to the uterus, with minimal bleeding.

## Data Availability

Not applicable.

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
