# Peer review of "Biologically-Based Notions About Uterine Bleeding During Myomectomy: Reasoning on Tradition and New Concepts"

_medsci, 2025, doi:10.3390/medsci13020068_

Round 1
Reviewer 1 Report
Comments and Suggestions for Authors
This article is a review of traditional and current methods of surgical management for uterine fibroids. The following are suggested to further improve this paper:
- In the introduction, please add the aims/objectives of the paper.
- Lines 118-119: Please clarify what the authors mean by "rising incidence of infertility...in older patients." Avoid using older and use age ranges. Older patients may include post-menopausal women, who are physiologically not reproductive.
- Line 145: Please avoid starting a paragraph with "It." Please re-instate what the "it" refers to. Add references to this paragraph, as well.
- Lines 155-157: These two sentences can be added to the prior paragraph.
- Lines 163-164: What does it mean by "normal blood volume"? Does this refer to the range of reported blood loss after myomectomy?
- Liness 277-282: Please define low-quality and moderate-quality evidence. Add this information into the "Introduction," if applicable.
- Lines 285-293: Please add references.
- Figure 5: Please add label into the figure to demonstrate vascular ring of pseudocapsule.
- Be consistent with number of decimal places.
- Multiple grammatical and spelling errors.
Author Response
This article is a review of traditional and current methods of surgical management for uterine fibroids. The following are suggested to further improve this paper:
In the introduction, please add the aims/objectives of the paper.
Answer: in accordance with the reviewer's right request, we have totally reviewed the introduction adding the aims/objectives of the paper.
Lines 118-119: Please clarify what the authors mean by "rising incidence of infertility...in older patients." Avoid using older and use age ranges. Older patients may include post-menopausal women, who are physiologically not reproductive.
Answer: we agree with the reviewer, we changed the sentence, making it clearer and adding a reference.
Line 145: Please avoid starting a paragraph with "It." Please re-instate what the "it" refers to. Add references to this paragraph, as well.
Answer: We have correctly modified the sentence, adding the reference.
Lines 155-157: These two sentences can be added to the prior paragraph.
Answer: We have inserted the two sentences into the prior paragraph.
Lines 163-164: What does it mean by "normal blood volume"? Does this refer to the range of reported blood loss after myomectomy?
Answer: we have rightly changed the meaning of the sentence, bringing back into the sentence “the blood loss after a myomectomy”.
Lines 277-282: Please define low-quality and moderate-quality evidence. Add this information into the "Introduction," if applicable.
Answer: Kongnyuy EJ and Wiysonge CS, in their paper entitled: “Interventions to reduce hemorrhage during myomectomy for fibroids”, published on Cochrane Database Syst Rev. 2014 Aug 15;2014(8):CD005355, defined the GRADE of quality evidence, defining four levels of evidence quality: high, moderate, low, and very low. We are not moving this sentence because, after consultation, we have decided that this sentence is best left in the paragraph: Traditional techniques to reduce bleeding.
Lines 285-293: Please add references.
Answer: We added the missing reference (by our mistake), but we left the Reference of Dr. De Falco, as he correctly studied the use of GnRH analogues in myomectomy with the effects on the pseudocapsule.
Figure 5: Please add label into the figure to demonstrate vascular ring of pseudocapsule.
Answer: We have corrected the label explaining the figure, otherwise the figure is altered by the added label.
Be consistent with number of decimal places.
Answer: We have revised the text, trying to correct all the reported errors, including number of decimal places.
Multiple grammatical and spelling errors.
Answer: The text has been reviewed and corrected by a native English Speaker.
Reviewer 2 Report
Comments and Suggestions for Authors
This manuscript explores biologically-based strategies for managing uterine bleeding during myomectomy, with a special focus on the intracapsular myomectomy technique. The topic is clinically important and aligns well with the current shift toward fertility-preserving surgeries. However, the manuscript, in its current form, requires major revisions before it can be considered for publication.
The main comments are as follows:
1) Structural and Logical Organization: The manuscript is excessively lengthy and repetitive in multiple sections (especially Sections 5–7); There is a lack of clear hierarchical structure (e.g., multiple subtopics are nested without clear transitions).
2) Redundancy and Wordiness: Key concepts (e.g., the anatomy of the myoma pseudocapsule, importance of preserving neurovascular bundles) are repeated almost verbatim across several sections.
3) Overemphasis on Intracapsular Myomectomy: While the focus on intracapsular myomectomy is understandable, the manuscript portrays it as universally superior without sufficiently discussing potential limitations, such as surgical learning curve, inapplicability in degenerated fibroids, or very large fibroids.
4) Insufficient Critical Appraisal of Cited Literature: The paper often cites studies in a descriptive manner without critical evaluation of study quality, sample sizes, biases, or contradictions between different studies.
5) Abstract is too long and unfocused. Please summarize key findings and conclusions more succinctly (within 250-300 words).
6) In Section "introduction", authors should explicitly state what previous reviews lack and what this review aims to add.
In summary, this MS addresses an important clinical challenge and has the potential to contribute meaningfully to the field. However, substantial restructuring, content condensation, critical analysis enhancement, and English polishing are needed before further consideration.
Author Response
The main comments are as follows:
Structural and Logical Organization: The manuscript is excessively lengthy and repetitive in multiple sections (especially Sections 5–7); There is a lack of clear hierarchical structure (e.g., multiple subtopics are nested without clear transitions).
Answer: Although we agree with the reviewer, the issue was too lengthy and complex to be condensed into a shorter piece because it is still being addressed in several papers and at numerous scientific conferences and congresses. We have attempted to report on current scientific research and emerging concepts in the text, attempting to address some of the current debates in the field of myomectomy. Paragraphs 5-7 focus on the most common methods of reducing surgical bleeding during myomectomy and we cannot think of reducing them, as the scientific literature is too rich in these papers to ignore. In any case, in the revision of the text, we tried to reduce some sentences, condensing some concepts, as rightly recommended by the reviewer.
Redundancy and Wordiness: Key concepts (e.g., the anatomy of the myoma pseudocapsule, importance of preserving neurovascular bundles) are repeated almost verbatim across several sections.
Answer: we agree with reviewer and in revising the text, we remodeled some sentences, leaving the concepts intact and reducing scientific redundancies. However, after having illustrated the anatomy of the fibroma, we have reworked paragraph 8 in some points, which is essential to explain in detail the reasons for the success of intracapsular surgery, in terms of less surgical bleeding and improved muscle healing.
Overemphasis on Intracapsular Myomectomy: While the focus on intracapsular myomectomy is understandable, the manuscript portrays it as universally superior without sufficiently discussing potential limitations, such as surgical learning curve, inapplicability in degenerated fibroids, or very large fibroids.
Answer: as surgeons, we firmly believe that intracapsular myomectomy is the most effective surgical approach for both open and endoscopic procedures. In contrast to existing approaches, which primarily address ways to lower blood flow to the uterus and, consequently, to the fibroid, the manuscript actually addresses the advantages of this physiologically based strategy. This is one of the subjects covered in the paper, and while it is lengthy (we concur with the reviewer), it covers all the literature before explaining why the intracapsular approach should be used for every myomectomy. We believe that the intracapsular myomectomy technique has no potential limitations because its surgical learning curve is the same as that of a normal myomectomy (only with the foresight to immediately identify the surface of the fibroma from its pseudocapsule, from which it must be separated), it can be applied to very large fibroids (which have already been published) using the open technique, and it can even be used to enucleate degenerating fibroids in open technique (since the surgeon can better manipulate the fibroid with his fingers during its enucleation, unlike laparoscopy where he is forced to apply forceps that tear the tissue).
Insufficient Critical Appraisal of Cited Literature: The paper often cites studies in a descriptive manner without critical evaluation of study quality, sample sizes, biases, or contradictions between different studies.
Answer: It is unquestionably true that the article describes the studies it cites without critically assessing the study's quality, sample size, or any inconsistencies or distortions among the several studies. However, this is not our goal at all (we specified it in the introduction). All the material that we have quoted has been done so solely to outline the current practices in operating rooms and wards for patients having myomectomy. Our issue is biological in nature rather than pharmacological or vascular. Since all the techniques are designed to decrease the flow to the uterus and, consequently, to the fibroid, they are theoretically incorrect. Based on the anatomy and pathology of the fibroid and uterine biology, we have described the surgical principles. We have already gone beyond the discussion of the quality of the studies in the literature, and we have presented a critical analysis of the need to look "beyond" the vascular tree that supplies the uterus as a target for mechanical and chemical approaches. A myomectomy should not be viewed in the current manner described in literature; rather, it should be viewed in terms of the biological preservation of the uterus for a future pregnancy, the healing process, and consequently the "reasoned" bleeding that occurs during and after the procedure without causing anatomical harm to the uterus and its surrounding tissues.
Abstract is too long and unfocused. Please summarize key findings and conclusions more succinctly (within 250-300 words).
Answer: In agreement with the reviewer, we have reduced the length of the abstract.
In Section "introduction", authors should explicitly state what previous reviews lack and what this review aims to add.
Answer: We have made changes to the manuscript's introduction section in accordance with the reviewer's suggestions, attempting to clarify our intentions for the article's structure and goals.
In summary, this MS addresses an important clinical challenge and has the potential to contribute meaningfully to the field. However, substantial restructuring, content condensation, critical analysis enhancement, and English polishing are needed before further consideration.
Answer: We agree with the reviewer, for this reason we have further revised the manuscript in order to satisfy the reviewer's requests and make the manuscript more feasible and readable, trying to respect the requests of the other reviewers too.
Reviewer 3 Report
Comments and Suggestions for Authors
The authors explain the importance of myomectomy as a conservative operation, emphasizing the complexity of intraoperative hemorrhage control. They propose a reevaluation of bleeding reduction techniques through a biological perspective on the fibrovascular pseudocapsule of the myoma. Key ideas from the manuscript: 1. Laparoscopy is superior to laparotomy in terms of blood loss, with no difference in complication rates. 2. Incorrect identification of the cleavage plane causes significant hemorrhage. The pseudocapsule (formed by compression of the myometrium around the tumor) contains neuropeptides and angiogenic factors, an important aspect in healing. Pharmacological options for reducing bleeding are widely discussed, as are mechanical options. However, intracapsular myomectomy represents the biological approach. This is the essence of the innovation brought by the article. The authors advocate respecting the cleavage plane between the myoma and the pseudocapsule, without using pharmacological or mechanical methods. The manuscript introduces a new biological concept for surgical practice. It has scientific rigor, clinical applicability. Limitations of the study: generalization of the proposal of intracapsular myomectomy as the "gold standard" requires additional validation through prospective multicenter studies. Value of the study: it is essential reading for gynecologists
Author Response
The authors explain the importance of myomectomy as a conservative operation, emphasizing the complexity of intraoperative hemorrhage control. They propose a reevaluation of bleeding reduction techniques through a biological perspective on the fibrovascular pseudocapsule of the myoma.
Key ideas from the manuscript: 1. Laparoscopy is superior to laparotomy in terms of blood loss, with no difference in complication rates. 2. Incorrect identification of the cleavage plane causes significant hemorrhage. The pseudocapsule (formed by compression of the myometrium around the tumor) contains neuropeptides and angiogenic factors, an important aspect in healing. Pharmacological options for reducing bleeding are widely discussed, as are mechanical options. However, intracapsular myomectomy represents the biological approach. This is the essence of the innovation brought by the article.
The authors advocate respecting the cleavage plane between the myoma and the pseudocapsule, without using pharmacological or mechanical methods. The manuscript introduces a new biological concept for surgical practice. It has scientific rigor, clinical applicability.
Limitations of the study: generalization of the proposal of intracapsular myomectomy as the "gold standard" requires additional validation through prospective multicenter studies. Value of the study: it is essential reading for gynecologists.
Answer: We thank the reviewer for his positive evaluation of the paper. We absolutely agree that, as a limitation of the study, there is that the intracapsular myomectomy as the "gold standard" requires additional validation through prospective multicenter studies. For this reason, we have inserted this sentence in the paragraph before the conclusions.
Reviewer 4 Report
Comments and Suggestions for Authors
Thank you for the opportunity to review the manuscript "BIOLOGICALLY-BASED NOTIONS ABOUT UTERINE BLEEDING DURING MYOMECTOMY: REASONING ON TRADITION AND NEW CONCEPTS."
It is a well written narrative review of the literature, about the aspects of minimizing bleeding during myomectomy. The authors describe the already known technique that surgeons must find and dissect-remove the myoma from its pseudocapsule with minimal and coagulation.
It should be stated in the title and in the text of the manuscript (Abstract, Methods, Conclusions) that this is a narrative review of the literature .
Author Response
Thank you for the opportunity to review the manuscript "BIOLOGICALLY-BASED NOTIONS ABOUT UTERINE BLEEDING DURING MYOMECTOMY: REASONING ON TRADITION AND NEW CONCEPTS."
It is a well written narrative review of the literature, about the aspects of minimizing bleeding during myomectomy. The authors describe the already known technique that surgeons must find and dissect-remove the myoma from its pseudocapsule with minimal and coagulation.
It should be stated in the title and in the text of the manuscript (Abstract, Methods, Conclusions) that this is a narrative review of the literature.
Answer: We thank the reviewer for his positive evaluation of the paper. We strongly agree that this is a narrative review and, as requested by the reviewer, we have highlighted this in the abstract, in the introduction to the M&Ms and in the conclusion of the paper. Furthermore, it has been highlighted in the header of the paper, above the title.
Round 2
Reviewer 2 Report
Comments and Suggestions for Authors
No comments.
Author Response
We thank the reviewer for his positive evaluation of the paper.